# Examining Predictors and Outcomes of Decent Work among Korean Workers

**DOI:** 10.3390/ijerph19031100

**Published:** 2022-01-19

**Authors:** Minsun Kim, Jaehoon Kim

**Affiliations:** 1Department of Psychology and Psychotherapy, Dankook University, Cheonan 31116, Korea; 2Department of Counseling Psychology, Global Cyber University, Seoul 06022, Korea; jaekim82@gmail.com

**Keywords:** psychology of working framework/theory, economic constraints, social marginalization, work volition, career adaptability, decent work, job satisfaction, life satisfaction

## Abstract

The goal of the present study is to examine the psychology of working framework/theory with a sample of Korean workers. This study examined the structural model of sociocultural factors (i.e., economic constraints and social marginalization), psychological variables (i.e., work volition and career adaptability), and outcomes of decent work based on the psychology of working framework. This study assumed that decent work helps all workers attain a sense of self-respect, dignity, experience freedom and security in the work environment and provides an opportunity for workers to contribute to society. Data were collected from 420 Korean workers, with an average age of 39.13 years (SD = 9.26). We used a hypothesis model that did not assume a direct path from economic constraints and social marginalization to decent work and work volition and career adaptation to job satisfaction and life satisfaction. We also employed an alternative model that assumed all of its paths and compared the models’ goodness of fit based on prior studies. Results indicated that alternative models have higher goodness of fit than hypothesis models. All path coefficients were significant except for the direct path from social marginalization to work volition and career adaptability to life satisfaction. Additionally, work volition and career adaptability mediated both the relationship between social marginalization and job satisfaction and between marginalization and life satisfaction. This study enabled the comprehensive examination of the relevance of various social environments and psychological and occupational characteristics that should be considered when exploring job or life satisfaction in the process of career counseling.

## 1. Examining Predictors and Outcomes of Decent Work among Korean Workers

Work is a critical component of life and a means of expressing one’s self-concept. People maintain their self-esteem and social relations through work and improve their physical and mental health [1,2,3,4]. However, job security has rapidly declined due to the global COVID-19 pandemic. As of April 2020, 470,000 jobs have disappeared compared to the previous year, and 1.5 million workers reported being on temporary layoff [5].

In Korea, the number of people leaving work voluntarily or involuntarily due to the pandemic has also increased [6]. The current unemployment rate among all age groups, including young people, has risen rapidly. As of January 2021, 2,196,000 people were involuntarily unemployed. The most common reason cited for this was “completion of temporary or seasonal work” (50.3%), followed by “lack of work or poor business” (22.1%), “honorary retirement, layoffs” (15.8%), and “workplace closures” (11.8%). Additionally, before their unemployment, temporary and daily workers accounted for more than 60% of the total workforce—40.3% and 23.2%, respectively—while full-time workers accounted for 18.2%.

Job stability is a pertinent factor in determining sustainability and predictability in an individual’s life [4]. Moreover, unstable employment not only causes emotional difficulties for individuals but can also be linked to social problems, such as family breakdown, child abuse, and violence [7,8,9,10,11]. Notably, social justice counseling is often emphasized, and it has been argued that clients’ social and environmental characteristics must be understood along with their psychological characteristics [1,12]. Lee, Seo, and Kim [13] argued that it is the psychological aspects of job stress or career troubles that clients share during counseling and their actual experiences, such as oppression and discrimination. Therefore, this study examined the applicability of the PWT [14] in Korea; this concept comprehensively explains the psychological, social, and environmental aspects that affect decent work and the results thereof. Decent work includes individuals’ subjective perceptions, such as the appropriate compensation for work, feeling stable in social relationships, proper time and rest, and consistency between family and work values [15]. Researchers in career psychology have also conceptualized aspects of decent work from a psychological perspective and are now interested in the influence of decent work on individuals [1]. In the last decade, Korean studies introduced the concept of decent work and factors that influence entry into decent work among young and middle-aged people, e.g., [16,17,18,19]. Such studies have explored domestic jobs’ quality and introduced difficulty of employment as a distinguishing characteristic of low-paying jobs, e.g., [20]. Additionally, the possibility of entering decent work was determined according to the workers’ academic background, whether they were engaged in full-time jobs, job search period, and company size. There is a growing interest in decent work in Korea, and most studies have focused on demographic characteristics and the previous job experience of individuals, which affect the quality of work. However, the comprehensive understanding of the psychological, social, and environmental characteristics that determine decent work from a psychological perspective remains limited.

PWT is based on the fact that, in real life, it is difficult for everyone to enjoy the freedom of choice in their career-related decision-making, and people may face various career-related limitations [21,22]. PWT focuses on contextual backgrounds to explain how psychological work factors affect an individual’s wellbeing [15]. Related studies have been conducted through models based on PWT, i.e., [22,23]. PWT can help analyze elements for understanding individuals’ wellbeing, including theoretical inclusiveness, social justice, personal level practice, and institutional and policy implications [24,25].

In PWT, individuals’ psychological satisfaction obtained through work is reflected on, and conceptualized as, a psychological factor. It emphasizes that individuals who experience many environmental constraints in their career selection may lack opportunities to choose their jobs freely, and environmental constraints should factor in constraints in career choice faced by a minority [15,26,27]. PWT assumes that psychological variables (e.g., work volition and career adaptability) mediate the relationship between environmental barriers and psychological outcomes. Specifically, the theory explains the influence of social structural factors on individual work and presents the predecessors and outcome factors of decent work, centering on this concept [2,15]. However, Korean studies related to PWT are still in their early stages [28,29]. In particular, most studies have established psychological factors as the mediator factors between social structural factors and one’s satisfaction [30]. However, having a decent work acts as an important factor in an individual’s life in that it affects the individual’s psychological wellbeing and job fullness [24]. In this study, based on the arguments of various scholars who state that work plays an essential role in an individual’s life [1,31], we tried to determine how decent work affects outcome factors.

### 1.1. Purpose of the Study

We sought to extend prior research on the PWT framework/theory by examining its potential to shed light on South Korean workers’ job and life satisfaction. Based on the PWT models and findings of recent studies, we formulated several sets of predictions, as illustrated in the path diagram in Figure 1.

PWT sparked psychologists’ interest in terms of the psychological functions of a decent works for an individual. Decent work not only improves an individual’s job satisfaction but also plays an important role in enhancing wellbeing by meeting basic needs (e.g., survival, self-determination, social connection). Due to the prolonged economic deterioration accompanied by COVID-19, South Korea is experiencing increased socioeconomic instability and lower employment stability, as well as job polarization [6]. According to Statistics Korea, in 2020, the employment rate of the youth fell 2.5% from the previous year to 41.3%, and the number of employed people in their 20s and 30s fell sharply; it is predicted that the unemployment issue will be difficult to resolve in a short space of time. By failing to satisfy fundamental needs in such a situation, the loss of work undermines individual and societal wellbeing in the long run, and employment instability goes beyond individual problems and can lead to family and social issues.

Through this study, we will be able to comprehensively understand individual background variables, psychological variables, and job quality that affect the life and job satisfaction of Korean workers and examine the applicability of the PTW developed in the United States. PWT research in the past has focused more on exploring previous studies of decent work. This study can provide an opportunity to overcome the limitations of career psychology in Asian countries such as Korea, which predicts individual job and life satisfaction-based decent work. Additionally, it will be a foundational study to emphasize the importance of career choice and job quality in society at a time when the polarization of social classes is deepening.

### 1.2. Hypothesized

#### 1.2.1. Economic Constraints and Social Marginalization

Race/ethnicity, social class, gender, and other contextual variables are the principal factors that determine an individual’s social status and level of marginalization [32]. In PWT, economic constraints and social marginalization are structural and environmental factors influencing decent work [15,33]. Furthermore, economic difficulties and social marginalization are examined as critical sociostructural factors that affect work volition, career adaptability, and the approach to decent work [15,33].

This study explored economic difficulties as an antecedent variable for decent work. A Korean study found that the working poor who experience financial difficulties have a very low chance of obtaining, or changing to, a decent job [16]. In other words, if people are in an economically difficult situation, people are more likely to obtain a part-time or general labor job, choosing jobs that are easily accessible over a quality job, which was found to be a factor that lowers the possibility of moving to a decent job. A study on college graduates revealed that the more economically stable the home, the more time and financial support are given to children to prepare for job search [17]. Previous studies have indicated that students perceived a limitation in career choice due to economic difficulties, despite being highly career-motivated [33,34,35,36]. Additionally, some studies, i.e., [37,38], have shown that economic constraints and difficulties can lower work volition and career adaptability. In other words, individuals’ socioeconomic levels and background factors are closely related to career choices.

Social marginalization refers to the perception that individuals (or groups of individuals) are in a position without power in society, and the experience of social marginalization is a significant barrier to obtaining stable jobs. In PWT, social class is a representative factor in determining social marginalization [15,39]. Autin et al.’s [40] study of college students showed that lower perceived social class negatively affects work volition and career adaptability. In Korea, perceived social class is positively linked to career adaptability [28]. The results showed that perceived constraints, such as social class and academic background, highly predicted work volition and career adaptability. Taken together, economic constraints and social marginalization are hypothesized to be positively associated with work volition and negatively associated with career adaptability.

Hypotheses 1–4: Economic constraints and social marginalization relate to work volition (Hypotheses 1 and 3) and career adaptability (Hypotheses 2 and 4).

**Hypothesis** **1** **(H1).**
*Economic constraints would be negatively to work volition.*


**Hypothesis** **2** **(H2).**
*Economic constraints would be negatively to career adaptability.*


**Hypothesis** **3** **(H3).**
*Social marginalization would be negatively to work volition.*


**Hypothesis** **4** **(H4).**
*Social marginalization would be negatively to career adaptability.*


#### 1.2.2. Work Volition and Career Adaptability

Work volition and career adaptability are psychological factors among the four preceding factors (economic constraints, social marginalization, work volition, and career adaptability) that influence obtaining decent work. They are initial mediators that explain why it is difficult for financially constrained and discriminated against people to obtain decent work [15]. Work volition is the ability to choose one’s job despite environmental constraints [41]. It was confirmed as a psychological factor that positively affects individuals. According to previous studies, work volition enhances university students’ academic satisfaction and career decision-making self-efficacy [42]. Studies have also shown that when workers have high work volition, their work and job commitment are high [33], and they can procure decent work [40].

Career adaptability is an individual’s flexibility to cope with various needs in the job field [43,44]. It is a multidimensional concept that includes concern, control, curiosity, and confidence [43]. It is positively associated with decent work [28,38,45]. Targeting 315 adult workers in Korea, An [28] aimed at exploring the main contextual factors that limit the career development of Korean adults. The results showed that the work volition and career adaptability of adult office workers positively predicted decent work. Based on this, the following hypotheses were established.

Hypotheses 5 and 6: Work volition (Hypothesis 5) and career adaptability (Hypothesis 6) positively related to decent work.

**Hypothesis** **5** **(H5).**
*Work volition would be positively to decent work.*


**Hypothesis** **6** **(H6).**
*Career adaptability would be positively to decent work.*


#### 1.2.3. Life and Job Satisfaction Are Outcomes of Decent Work

In previous studies, e.g., [46], decent work was found to be positively linked to job satisfaction. It has also been found to be related to overall satisfaction and positive emotions in individuals’ lives [47]. To measure psychological wellbeing and work fulfillment, life and job satisfaction were examined [15].

Although substantial research based on Western contexts has investigated PWT concerning workers’ job and life satisfaction, relatively few empirical studies with Asian samples [38] have been published in major journals in the field of vocational psychology. While the abovementioned studies demonstrated that the PWT model is applicable in Asian cultures, Ma et al. [38] showed that some mediation effects differed from the hypothesis of the PWT model. Specifically, Ma et al.’s [38] study on Chinese university students found that the overall indirect effect mediated by work volition, career adaptability, and decent work was not significant in the relationship between economic difficulties and career exploration. These results imply that there are limitations to indirectly applying the PWT model to Asian cultures. Additionally, past PWT research has focused more on exploring previous studies on decent work. Thus, there remains a dearth of research on how decent work affects outcome factors [48]. A research hypothesis was established based on these previous studies. Korea continues to be one of the OECD countries with the most extended working hours [49]. Unlike other countries, before Korea’s employment relationship or corporate culture could adequately mature, it had achieved rapid economic development centering on conglomerates; the business structure of subcontractors had created poor employment conditions for small-to-medium-sized businesses [50]. Korean corporate culture discourages employees from leaving work at a fixed time or from using vacation days, and employees are apprehensive about leaving work before their superiors. Additionally, the overall lack of infrastructure for leisure negatively affects the proliferation of decent work. Before discussing decent work in Korea, the government handled the problem by reducing work hours, which developed into a time-selective job policy. South Korea’s Ministry of Employment and Labor defines part-time work as “work that meets the voluntary needs of workers, guarantees basic working conditions, and does not unreasonably discriminate in terms of working conditions”. The Ministry states that workers can choose to become a part-timer for (1) work and life balance, (2) gradual retirement and re-employment, (3) working and studying at the same time, and so on in consultation with the business owner regarding working hours, start and end times of work, working days, and so on. In short, it can be understood as work that the worker chooses to perform part-time for work–life balance. However, Korea’s legal bases for time-selective jobs are weaker, and equal treatment and antidiscrimination measures are inadequate compared to those in other developed countries. Several mounting criticisms claim that such a policy leads to difficulties in switching to full-time positions [51,52] and the marginalization of female workers [53].

**Hypothesis** **7** **(H7).**
*Decent work would be positively related to job satisfaction.*


**Hypothesis** **8** **(H8).**
*Decent work would be positively related to life satisfaction.*


#### 1.2.4. Economic Constraints, Social Marginalization, and Decent Work

We have described a hypothesized model (Figure 1) that appears theoretically sound, logically consistent, and based on prior research. However, it is possible that other models could explain the relationships among hypothesized variables. Therefore, this study attempted to adopt a better model by establishing an alternative model and a research model that also examines the above hypotheses. In Korea, perceived social class and perceived social marginalization were found to be positively linked to career adaptability and decent work [28].

Hypotheses 9 and 10: Economic constraints (Hypotheses 9) and social marginalization (Hypotheses 10) are negatively related to decent work.

**Hypothesis** **9** **(H9).**
*Economic constraints would be negatively to decent work.*


**Hypothesis** **10** **(H10).**
*Social marginalization would be negatively to decent work.*


#### 1.2.5. Relationship between Job Satisfaction and Life Satisfaction with Mediators

Work volition has a positive effect on positive career outcomes [54,55], positive emotions [15], life satisfaction [46,56], academic satisfaction [42], and job satisfaction [57]. For example, Duffy et al. [21] found that work volition was positively associated with job satisfaction among workers.

Hypotheses 11 and 12: Work volition is positively related to job satisfaction (Hypothesis 11) and life satisfaction (Hypothesis 12). 

**Hypothesis** **11** **(H11).**
*Work volition would be positively to job satisfaction.*


**Hypothesis** **12** **(H12).**
*Work volition would be positively to life satisfaction.*


Career adaptability was also related to job and life satisfaction [58]. The relationship between career adaptability and job satisfaction has been verified in various cultures. For instance, Kozan et al. [39] applied the PWT model to low-income individuals in Turkey and verified that career adaptability significantly positively affects job and life satisfaction.

Hypotheses 13 and 14: Career adaptability is positively related to job satisfaction (Hypothesis 13) and life satisfaction (Hypothesis 14).

**Hypothesis** **13** **(H13).**
*Career adaptability would be positively to job satisfaction.*


**Hypothesis** **14** **(H14).**
*Career adaptability would be positively to life satisfaction.*


Although studies that have been conducted in countries other than the United States are limited, Wang et al. [59] verified the applicability of PWT among Chinese adults. Consequently, the model that mediates the relationship between work volition, subjective social status, job satisfaction, and career persistence was found to explain the data gathered from these Chinese adults appropriately. Regarding path coefficients, work volition and subjective social status had a significant positive effect on work volition, associating life satisfaction and career persistence. In other words, contextual constraints, as assumed in PWT, limit the career resources of individuals, and lower their work volition [15]. Accordingly, we proposed that work volition and career adaptability mediate the link between economic constraints and social marginalization with job and life satisfaction.

This study sought to extend prior research on the PWT model and comprehensively understand the individual background variables, psychological variables, and job quality that affect life and job satisfaction among Korean workers. Recently, experts in the field of vocational psychology have conceptualized the subjective aspect of decent jobs from a psychological point of view and have begun to pay attention to the impact of decent work. Increasing access to predictable and secure jobs not only positively affects individuals’ mental and physical health, but also significantly affects social solidarity. In this context, psychologists also need to pay attention to how to define and measure decent work and identify environmental and psychological factors that allow people to access decent work or barriers that make it difficult to do so. Notwithstanding the global trend, in the field of vocational psychology in Korea, the research focus is mostly on examining the effect of individuals’ psychological characteristics on job and life satisfaction (e.g., match between personality and work environment) rather than the benefits decent jobs can bring to them. This is a limitation in the sense that it not only keeps the vocational psychologists from considering the impact of context and environment to which individuals belong, but also fails to reflect the research results showing that job quality is an important factor that determines individual job and life satisfaction.

Based on the PWT models and findings of recent studies, we formulated several sets of predictions, as illustrated in the path diagram in Figure 1 as the hypothesized model.

## 2. Method

### 2.1. Participants and Procedure

Data from 420 (213 women, 207 men) workers with a mean age of 39.09 years (SD = 9.30) were collected for this study. Of these participants, 177 (42.1%) were unmarried, and 241 (57.4%) were married. A large portion of the sample reported that they were college graduates (67.1%). In terms of years of work, 122 (29.0%) participants had fewer than 3 years, 123 (29.3%) had between 3 and 10 years, and 173 (41.2%) had more than 10 years. As of 2020, the average age of Korean workers was 42.9 years old, and the average age of the study participants was slightly younger. A total of 36.9% of the participants earned between KRW 2.01 million and 4 million, which indicates that a large number earned an average wage, considering that the average wage in 2019 was KRW 3.01 million. The demographic characteristics of the participants are presented in Table 1.

The survey was conducted online from October to November 2019, bearing in mind the ease of access for participants, the convenience of the questionnaires, and the diversity of sampling, as it was a nationwide survey [60]. Macromill Embrain, a specialized research agency, was recruited to conduct a survey on adults between 25 and 60 years of age residing in South Korea. This online research agency has a panel of about 1.3 million people including adults of various ages. As the subjects were adults with a job, a screening questionnaire included whether they currently have a full-time job, and if they answered they do not, the survey was ended. To recruit participants, the survey was posted on the Macromill Embrain homepage (https://www.panel.co.kr/user/main, accessed on 1 March 2019) and the qualified participants were allowed to respond. A total of 429 people completed the questionnaire, and the data of nine people with several unanswered questions were deleted. The purpose of this study was explained to the participants, and the questionnaire was only administered to those who had agreed to participate in the study. Participants were also provided with modest compensation (USD 8.95). They provided online signatures indicating that they had agreed to participate in this study.

### 2.2. Instruments

#### 2.2.1. Economic Constraints

In this study, economic constraints [61] were used to measure economic difficulties. Economic difficulties were measured using 12 items (e.g., lack of income, lack of money, family members arguing or having trouble with others, and delaying the purchase of household goods to save money), which were evaluated as a single factor on 5-point Likert-type scale (1 = strongly agree, 5 = strongly disagree). Higher scores on the scale indicate higher perceived economic constraints. Kang [61] found that economic constraints were positively correlated with subjectively perceived economic status and negatively correlated with total household income. In a study by Kang [61], Cronbach’s alpha was 0.88, while the present sample produced a Cronbach’s alpha of 0.86.

#### 2.2.2. Social Marginalization

In this study, four items were used to measure social marginalization. These were based on items used in the German GMF survey [62] to measure the institutional and sociostructural dimensions of disintegration. In the study by Issmer and Wagner [63], these questions were used to measure social marginalization. Social marginalization comprises four questions (e.g., “Problems will arise when finding a job because of my background”; “It is difficult for a person like me to live a normal life”). Responses were indicated on a 4-point Likert-type scale ranging from 1 (do not agree at all) to 4 (fully agree). Higher scores on the scale indicate higher social marginalization. For the translation of the scale, the first author, who holds a doctorate in counseling psychology, discussed the measures and translated them from English into Korean. Then, back-translation from Korean to English was conducted by a bilingual student (a psychology major) unfamiliar with both the original versions of the measures and the purpose of this study. Subsequently, a native English speaker with a master’s degree in counseling compared the original items with the back-translated items to evaluate semantic equivalence and accuracy. In Issmer and Wagner’s [63] study, Cronbach’s alpha for work volition was 0.76–0.83, while the present sample produced a Cronbach’s alpha of 0.82.

#### 2.2.3. Work Volition

In this study, the Work Volition Scale (WVS) developed by Duffy et al. [41] was used to measure work volition. It includes 14 questions that are classified into three factors: Volition (e.g., “I could choose the job I wanted”, “I feel complete control over my job choice”), financial constraints (e.g., “I will do whatever I can get when I get a job”, “I often have jobs I do not enjoy to support my family”), and structural constraints (e.g., “The current economic situation hinders me from having the job I want”, “Negative factors outside my personal control have had a great impact on my current career choice”). Each item was measured on a 7-point Likert-type scale (1 = strongly disagree, 7 = strongly agree). Higher scores on the scale mean higher work volition; average scores of the items belonging to financial constraints and structural constraints were used. In Duffy et al.’s [41] study, the Cronbach’s alphas for work volition, economic constraint, and structural constraint were 0.69, 0.78, and 0.64, respectively. Work volition showed significant negative correlations with control in work and neurosis, while it was positively correlated with extroversion and job satisfaction [41]. The present sample produced a Cronbach’s alpha of 0.75.

#### 2.2.4. Career Adaptability

Career adaptability was measured using the Career Adapt-Abilities Scale (CAAS) developed by Savickas and Profeli [43]. The scale was used to identify the homogeneity of the composition concept, reliability, and measurement in 13 countries. Tak [64] conducted a study on its utilization in Korea. It contains four subscales: concern, control, curiosity, and confidence. Each item was measured on a 5-point Likert-type scale (1 = strongly disagree, 5 = strongly agree). Higher scores on the scale indicate career adaptability. The reliability for the total score for the CAAS-Korea was 0.93, which was higher than the reliability of each of the four subscales of concern 0.85, control 0.80, curiosity 0.82, and confidence 0.84. The present sample produced a Cronbach’s alpha of 0.95 in the study by Tak [64].

#### 2.2.5. Decent Work

In this study, the Decent Work Scale (DWS) [65], validated by Nam and Kim [66], was used to measure decent work for Korean people. DWS contains 15 items that evaluate physically and interpersonally safe working conditions (e.g., “I feel emotionally safe when interacting with people while working”), access to healthcare (e.g., “I am getting good quality health insurance”), adequate compensation (e.g., “I think my pay is less than my conditions and career”), hours that allow for free time and rest (e.g., “I do not have enough time for activities other than work”), and organizational values complement family and social values (e.g., “The values of my organization are consistent with my family values”). Each item was rated on a 7-point Likert-type scale (1 = strongly disagree, 7 = strongly agree). Higher scores on the scale indicate decent work; the average scores for items belonging to each sub-factor were used. Duffy et al. [65] found that decent work had a significant positive correlation with job satisfaction and work meaning. The Cronbach’s alphas for all items, safe working environment, health insurance accessibility, appropriate compensation, working hours, and the company’s values meeting family and social values were 0.86, 0.79, 0.97, 0.87, 0.87, and 0.95, respectively. Nam and Kim [66] showed that safe working environment was 0.80, health insurance accessibility 0.94, appropriate compensation 0.85, working hours 0.58, and company’s values meeting family and social values 0.91. The present sample produced a Cronbach’s alpha of 0.89.

#### 2.2.6. Job Satisfaction

The Minnesota Satisfaction Questionnaire was used to measure adult job satisfaction. It was developed by Weiss et al. [67] and validated by Park [68] in Korea. It comprises 20 questions, measured by dividing job satisfaction into intrinsic, extrinsic, and general satisfaction. Job satisfaction was rated on a 5-point Likert-type scale (1 = strongly disagree, 5 = strongly agree). High scores indicate high job satisfaction. In Lee and Lee [30], the Cronbach’s alpha was 0.85, while the present sample produced a Cronbach’s alpha coefficient of 0.94.

#### 2.2.7. Life Satisfaction

Life satisfaction was measured using the Satisfaction with Life Scale (SWLS) [69], translated into Korean by Cho and Cha [70]. The SWLS comprises five items regarding satisfaction with their life as subjectively evaluated by individuals, such as “I live in a similar way to what I usually want” and “My life situation is very good”. The life satisfaction scale was rated on a 7-point Likert-type scale (1 = strongly disagree, 7 = strongly agree). High scores indicate high life satisfaction, which applies to multiple roles. The present sample produced a Cronbach’s alpha coefficient of 0.86.

### 2.3. Data Analysis

In this study, SPSS 21.0 (IBM Corp., New York, NY, USA) and AMOS 21.0 (IBM Corp., Chicago, IL, USA) were used to test the mediating effects of the structural model’s fitness and mediation. In the case of the *χ*^2^ verification, the model was easily rejected because the contents of the null hypothesis were strict; the final model fit was selected by considering the fit index because the sample size could be significantly influenced. Economic constraints measure only one factor with a total of 12 items, so in this study, three-item-parceling was created and used based on the suggestions of Russell and colleagues [71]. As a result of checking the skewness and kurtosis values of the measurement variables of the package, no variables exceeded 2 and 7 points in skewness and kurtosis, respectively [72]. We used three indices to assess the measurement model’s goodness of fit: (a) the comparative fit index (CFI) ≥ 0.90, (b) the Tucker–Lewis index (TLI) ≥ 0.90, and (c) the root mean square error of approximation (RMSEA) ≤ 0.08 [73,74].

## 3. Results

### 3.1. Preliminary Analyses

In this study, age, education, and working period, which are demographic characteristics that can affect the results, were controlled. A correlation analysis was conducted, and the mean, standard deviation, kurtosis, and skewness of each variable were examined to understand the overall characteristics of the data before conducting the structural equation analysis.

Table 2 presents the correlation between measurement variables, the mean and standard deviation of each variable, and the degree of distortion and kurtosis. In the relationship between the latent variables, economic constraints showed a positive correlation with perceived marginalization (*r* = 0.56), and we found that work volition (*r* = 0.29), career adaptability (*r* = −0.12), decent work (*r* = −0.33), job satisfaction (*r* = −0.20), and life satisfaction (*r* = −0.23) were negatively correlated with economic constraints. Social marginalization was negatively correlated with work volition (*r* = 0.17), career adaptability (*r* = 0.37), decent work (*r* = −0.39), job satisfaction (*r* = −0.35), and life satisfaction (*r* = −0.29). Additionally, work volition was positively related to career adaptability (*r* = 0.35), decent work (*r* = 0.15), job satisfaction (*r* = 0.40), and life satisfaction (*r* = 0.36). Career adaptability also showed a significant positive correlation with decent work (*r* = 0.46), job satisfaction (*r* = 0.66), and life satisfaction (*r* = 0.43). Finally, decent work was positively related to job satisfaction (*r* = 0.69) and life satisfaction (*r* = 0.55).

The maximum likelihood estimation method used in this study assumes multivariate normality of data. The absolute values of the skewness and kurtosis of the measurement variables did not exceed 2 and 7, respectively, thus satisfying the multivariate normality [75].

### 3.2. Test of the Hypothesized Model

#### Testing the Hypothesized Structural Model

Results of the hypothesized model provided an acceptable fit to the data: *χ*^2^ (361, *N* = 420) = 965.761, CFI = 0.92, TLI = 0.91, RMSEA = 0.063 (90% confidence interval = 0.058–0.068).

The model fit of the alternative model was as follows: *χ*^2^ (355, *N* = 420) = 851.403, CFI = 0.93, TLI = 0.92, RMSEA = 0.058 (90% confidence interval = 0.053–0.063). This study set the correlation between errors between work volition and career adaptability (*r* = 0.52) as the mediating variable based on the PWT assumption. In addition, the relationship between life satisfaction and job satisfaction was assumed, or the correlation between error terms of latent variable was not indicated. Verification of the chi-square difference showed that the difference between the two models was significant (Δ*χ*^2^ = 114.358, *df* = 6, *p <* 0.001) (see Table 3). This study also considered the model fit to set the final model because the sample size affected the chi-square value. The model fit difference between the two models was significant, and the added path coefficients of the hypothesized model were significant. Therefore, the alternative model was chosen as the final model. 

Consistent with the hypotheses, economic constraints were related to career adaptability (*β* = 0.25, *p* < 0.001), work volition (*β* = 0.37, *p* < 0.01), and decent work (*β* = −0.33, *p* < 0.01) (see Figure 2). Social marginalization was negatively related to career adaptability (*β* = −0.59, *p* < 0.001) and decent work (*β* = −0.17, *p* < 0.05). In addition, work volition (*β* = 0.24, *p* < 0.001) and career adaptability (*β* = 0.40, *p* < 0.001) were positively related to decent work. Lastly, decent work was positively related to job satisfaction (*β* = 0.71, *p* < 0.001) and life satisfaction (*β* = 0.74, *p* < 0.001). However, Hypothesis 3 (social marginalization is negatively related to work volition) and Hypothesis 4 (career adaptability is directly related to life satisfaction) were not supported.

### 3.3. Assessment of Mediated Effects

Shrout and Bolger’s [76] Bootstrap Procedure Was Used to Estimate the Significance of the Indirect Effects. We instructed AMOS 21.0 to create 10,000 bootstrap samples from the original dataset (*N* = 420) by random sampling with replacement. Then, when analyzing the structural model, we generated indirect effects and bias-corrected confidence intervals (CIs) around the indirect effects. Indirect effects are deemed to be significant if the 95% CI does not include zero. The study verified a multiple-mediation model including several parameters. For the verification, the results are combined after analyzing all indirect effects that may exist between variables. The total indirect effect was estimated by adding up the five indirect effects, as follows, to calculate the indirect effect that exists between economic difficulties and job satisfaction. The estimation did not consider a direct effect, as a direct connection between the two variables was not assumed.
Total Indirect Effect: a + b + c + d + e = 0.024
a. (economic constraints × work volition × job satisfaction) = 0.070b. (economic constraints × career adaptability × job satisfaction) = 0.047c. (economic constraints × decent work × job satisfaction) = −0.227d. (economic constraints × work volition × decent work × job satisfaction) = 0.063e. (economic constraints × career adaptability × decent work × job satisfaction) = 0.071

Table 4 shows the estimates for the direct and indirect effects. Our hypothesis that work volition and career adaptability would significantly mediate the association between economic constraints and decent work (*β* = 0.19, *p <* 0.001) was supported. The direct and indirect effects of economic constraints to decent jobs were significant, while the total effect was not significant. Kim and Kim [77] argued that this phenomenon could occur if the direct and indirect effects were in different directions, or if the absolute value of the direct effects was greater than the total effects, which was named as an inconsistent mediation. Some researchers noted that, due to the low power of the total effect, it is important to note that the total effect is not significant, and in this case, there is also a mediated effect [78,79]. However, the indirect effects of psychological variables on the relationship between economic constraints, job satisfaction, and life satisfaction were insignificant. The mediated pathways from social marginalization through work volition and career adaptability to decent work (*β* = −0.25, *p <* 0.001) and social marginalization through work volition, career adaptability, decent work to job satisfaction (*β* = −0.42, *p <* 0.01), and life satisfaction (*β* = −0.33, *p <* 0.01) were significant.

## 4. Discussion

Based on PWT and previous studies, this study examined whether social marginalization and economic constraints affect an individual’s decent work, job satisfaction, and life satisfaction through work volition and career adaptability.

First, our results showed that the alternative structural model based on PWT offered a good fit for the data, suggesting that the PWT model can serve as a useful theoretical framework for explaining Korean workers’ job and life satisfaction. In addition to the assumptions of the existing PWT model, we added direct paths from economic constraints and social marginalization to decent work, and direct paths from work volition and career adaptability to work satisfaction and life satisfaction in the alternative model. Studies have shown that alternative models are more suitable than hypothetical models that do not assume direct paths of some variables, meaning that contextual variables can directly impact decent work for Koreans. It can also be seen that psychological aspects directly impact job satisfaction and life satisfaction, as well as indirect effects through decent work. The results suggest that PWT not only effectively explains the predictors and outcome variables of decent work studies, but may also have more direct relevance for some variables.

Examining the significant paths, economic constraints were significantly related to work volition and career adaptability. However, contrary to our hypothesis, higher economic constraints were associated with a greater likelihood of work volition and career adaptability. The present study suggests that having less access to economic resources means more work volition, which indicates that those who experience more economic constraints can cope more flexibly with changes to the occupational environment.

Considering the results of correlation analysis and previous studies, economic hardship can be interpreted as increasing the work volition while lowering career adaptability. Some studies in Korea have shown contrasting results that people who suffer economic hardships look and settle for menial or temporary jobs that can be easily found around them [12]. In contrast, people who have sufficient financial resources or have the support of their parents pass on a few job offers to find more decent works that are up to their standards. In other words, it may be recognized that the more difficult the economic hardship becomes, the more easily one is led into thinking that there are many jobs available for the money. In particular, if the socioeconomic status of the family is not high, the expectations of parents and neighbors for the individual’s career would not be high either, and the willingness to work for career choices as perceived by them can be high [78,80]. In fact, the study revealed that the more difficult the economic hardship is, the lower the perception on decent work. Therefore, in future studies, it would be necessary to expand the scope of research related to the willingness to work in reflection of the discrepancy between the individuals’ ideal perception of decent jobs and the realistic perception of jobs they can actually take. Therefore, the results of this study, which indicated that the higher the economic hardship, the higher the willingness to work, could be the manifestation of the current situation and cultural characteristics of Korean job seekers.

In addition, it is necessary to consider the influence of the scale when interpreting this study. Similar to this study, Ma et al. [38] also found that economic difficulties did not significantly affect work volition. Researchers mentioned the limitation of the scale that measures economic difficulties throughout life. As this study also used past-type questions (e.g., “there was a time when I couldn’t pay utility bills because I didn’t have money”) in measuring economic constraints, it may not have reflected the current economic difficulties. In order to more strictly interpret the relationship between the two variables, it will be necessary to reconsider the scale through follow-up studies.

On the other hand, it is necessary to consider statistical flaws and the appropriateness of the study model when discussing the results of economic hardship increasing career adaptability. According to the study by England et al. (2020) [81], the direction of the direct effect of the preceding variables on career adaptation (economic hardship, social marginalization) was derived differently from the hypothesis, and it rather showed the correlation that the higher the perception of economic hardship, the better the career adaptability. Upon examination, the author interpreted that in the process of analyzing the model, including the influence of the willingness to work, there may have been an indirect effect on career adaptation. Furthermore, the results suggested that career adaptability is not directly related to preceding variables that can be seen as environmental constraints, and that it is necessary to review the relationship between environmental constraints and career adaptability in the PWT model.

Second, social marginalization was negatively related to career adaptability and not related to work volition. Consistent with previous studies [22,40], perceived social marginalization can lower career adaptability. This is consistent with the assumption of PWT that the higher one’s autonomy in career choice is, the higher one’s adaptability in one’s career or field is [82]. However, contrary to previous studies, social marginalization was not related to work volition. Previous studies, i.e., [22,81], consistently perceived social marginalization as a factor that significantly lowers work volition.

Work volition positively affected decent work, job satisfaction, and life satisfaction. These results are consistent with the PWT model and previous findings that showed that work volition was related to decent work [81,82]. It means that if people think they have the freedom to choose work despite environmental constraints, they are more likely to procure decent work. The direct paths from work volition to job satisfaction and life satisfaction were statistically significant. These results are consistent with previous findings [39,56], which showed that work volition was related to life and job satisfaction, meaning that these can be low when people feel that they cannot freely choose jobs due to economic reasons, social class, environmental constraints, and so on.

Third, career adaptability positively affected decent work and job satisfaction. This finding is consistent with previous studies that showed that career adaptability is related to decent work [28,81] and job satisfaction [83,84]. This means that if individuals’ flexibility in coping with environmental constraints increases, their ability to access decent work can also increase [15]. Contrary to our hypothesis, the results indicated a lack of association between career adaptability and life satisfaction. This is inconsistent with the finding that career adaptability significantly affects life satisfaction [84]. In previous studies, there was a difference in the relationship between life satisfaction and career adaptability based on the sub-factors of career adaptability [85]. Interest and control had a significant relationship with life satisfaction, while curiosity and confidence did not. Savickas [44] stated that interest and control are the most critical sub-factors of career adaptability. It is necessary to further examine the differences according to the sub-factors of career adaptability through follow-up research.

Finally, decent work was related to job and life satisfaction. Previous studies have found that decent work is related to the job [55,86] and life satisfaction [43]. It means that constant efforts are needed to improve the quality of jobs by applying more multifaceted criteria for evaluating them, such as job compensation, stability of the working environment, work intensity, employment stability, and development potential [10]. Additionally, those who can work, but must work for a long time in a relatively poor or unsafe environment, can be linked to individual job satisfaction and personal life satisfaction [16]. In other words, the problem of ensuring job stability and expanding decent work can be linked to the right to pursue happiness as a member of society, which is beyond simply guaranteeing the right of individual labor. Further, the problem of labor needs to be examined based on life problems.

### 4.1. Implications and Recommendations for Career Counseling

This study is meaningful because it has examined the applicability of PWT, which has been widely used and evaluated in career psychology for the past five years among Korean workers. Through this, the results of previous studies and those of a study with Korean workers were directly compared, and suggestions for follow-up studies were proposed along with a discussion of the results. Particularly in Duffy et al.’s [15] psychological model, the hypothesis was verified through theoretical and empirical research; the empirical research was verified along with the paths, and it was established that it should be further expanded.

The practical implications of this study are as follows: first, for a socially marginal person, work volition and career adaptability decreased in this study. Thus, the more vulnerable people feel that they have limited options in choosing work or have difficulty flexibly coping with environmental changes [79,87,88]. Therefore, it is necessary to understand the difficulties of career decisions or adaptation to the occupational environment that a client reports in counseling, especially considering the environmental characteristics that pertain to the client. Additionally, to help clients cope with environmental constraints, it is necessary to think more widely about the alternatives available and lower their anxiety about the job change. 

Second, previous studies, e.g., [89], have shown that the probability of individuals having an excellent job after ten years increases if the characteristics of the first job are consistent with their college majors. Therefore, it is necessary to inform clients that continuous education and training are needed for stable jobs and help them plan their careers from a lifetime perspective. 

Third, the results of this study have shown that career adaptability is positively related to job satisfaction. In a study by Park and Lee [90], variables were divided into psychological, behavioral, and social environment variables related to career adaptability. A meta-analysis revealed that behavioral variables, such as strength utilization, accident coping skills, self-directed learning, and career preparation behavior were most likely to explain career adaptability. Therefore, it is necessary to improve the behavioral coping skills that clients can use through career counseling or educational programs.

Finally, the results of this study can be utilized to expand education programs for young people. The proportion of working youths has recently increased in Korea, and new problems have emerged due to changes in their labor environment and market. According to the results of the survey on the condition of young people’s part-time jobs [91,92], most perceived finding a job to be very difficult, and contracts for very short-term jobs with a high turnover rate were found to be scarce. There were many cases where less than the minimum wage was received; in many others, such as courier service delivery, the working environment’s safety was deficient because of high physical risks. The ratio of those working in occupational groups that did not receive industrial accident compensation even if an accident occurred was also high.

Furthermore, there were often infringements to relationship safety, with employers, guests, and colleagues being the main targets of insults, and profanity and verbal abuse were common and were found to be directed at the youngest employees. Vulnerable groups mainly showed a relatively high participation rate in the labor market compared to ordinary youths. This group was likely to be exposed to dangerous industries and rights violations. In Korea, the problem of part-time jobs for young people is now receiving society’s attention, and various policies have been proposed to guarantee their labor rights and actual condition surveys. Counselors should provide primary labor rights education to adolescents [93] and develop specialized education programs that fit the life characteristics of the youth in various vulnerable classes. It should also be mandatory for counselors to participate in professional education on the fundamental rights of youth labor and provide evaluation indicators and related guidelines on the quality of youth-friendly jobs at the central government level. There is also a need to expand formal institutions to help employers and employees actively cope with unfair treatment by providing education and various reporting channels to protect youth labor rights.

### 4.2. Limitations and Future Research Directions

This study has several limitations. First, this study sampled approximately 420 workers. About 70% of the participants were college graduates: relatively more highly educated were more likely to participate in this study. Nearly half were employees or sub-proximate employees. Therefore, in a follow-up study, the research should be expanded to include people of various academic backgrounds, careers, and age groups. Second, the study applied and evaluated the variables and hypotheses mentioned in the PWT [15], but it did not include personality variables of self-determination or intervention variables. Some paths (direct ones from social marginalization to work volition, and from career adaptability to life satisfaction) were not meaningful in the relationship of some variables, contrary to the existing hypothesis. Hence, it is necessary to analyze more deeply whether this path reflects cultural characteristics. Lastly, this study examined the influence of mediators, such as work volition and decent jobs. However, there are suggestions that, methodologically, when validating the path from mediators to dependent variables, it is necessary to assume the influence of several confounding variables on the dependent variables and use alternative analytical methods [94,95]. Therefore, it is necessary to consider the influence of confounding variables, such as gender, age, and personality, on the dependent variables in interpreting results. Subsequent studies need to use methods to minimize the influence of confounding variables in analyzing mediated effects. A domestic study based on the PWT assumed that factors such as academic background would have an important impact on the experience of social exclusion and the process of becoming marginalized. Achieving a good score on the university entrance exam is considered more important in South Korea than in any other country, and the socioeconomic status of parents was found to be a significant influence on children’s higher education and employment [28]. At a time when various social problems, such as rising house prices and the expansion and insecurity of nonregular employment, are increasing, the social polarization among young people is being mentioned as another social issue. Therefore, a follow-up study needs to more comprehensively include factors regarding social marginalization that may be experienced in South Korea.

## Figures and Tables

**Figure 1 ijerph-19-01100-f001:**
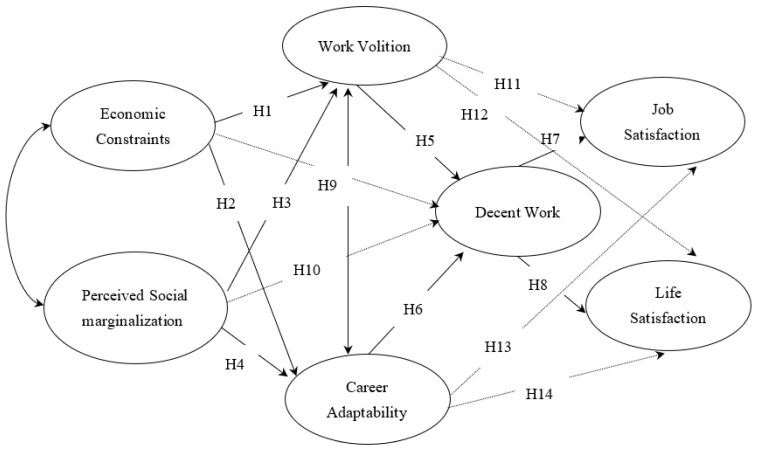
The hypothesized model. *Note.* Although not assumed in the PWT model, the paths added in this study are indicated by dashes. Age, education, and working period were controlled.

**Figure 2 ijerph-19-01100-f002:**
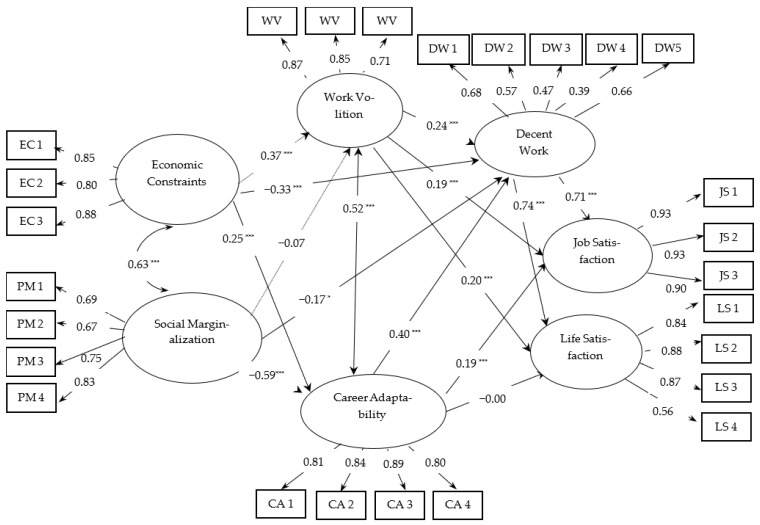
Standardized parameter estimates from the path analysis (*N =* 420). EC1–3 = item parceling economic constraints; PM1–4 = 4 items on social marginalization; WV1 = volition; WV2 = financial constraints; WV3 = structural constraints; CA1 = concern; CA2 = control; CA3 = curiosity; CA4 = confidence; DW1 = physically and interpersonally safe working conditions; DW2 = access to healthcare; DW3 = adequate compensation; DW4 = hours that allow for free time and rest; DW5 = organizational values complement family and social values; JS1 = intrinsic; JS2 = extrinsic; JS3 = general satisfaction; LS1–4 = 4 items on life satisfaction. Solid lines: significant paths, dashed line: nonsignificant paths. * *p* < 0.05; *** *p* < 0.001.

**Table 1 ijerph-19-01100-t001:** Demographic information.

Variable		Frequency	Percentage
Gender	Male	207	49.3
	Female	213	50.7
Marital Status	Single	177	42.1
	Married	243	57.9
Education	High school grade	84	20.0
	Bachelor’s degree	282	67.2
	Post graduate	54	12.8
Occupation	Office job	271	64.5
	Sales job	14	3.3
	Production worker	29	6.9
	Service job	78	18.6
	Professional job	28	6.7
Year of employment	>1	11	2.6
	1–3	112	26.7
	3–10	123	29.3
	10<	174	41.4
Job position	Staff	127	30.2
	Assistant manager	117	27.9
	Manager	123	29.3
	Director	53	12.6
Monthly household income	>KRW 2 million	41	9.8
	KRW 2.01–4 million	155	36.9
	KRW 4.01–6 million	107	25.5
	KRW 6.01–8 million	79	18.8
	KRW 8 million<	38	9.0

**Table 2 ijerph-19-01100-t002:** Descriptive statistics and bivariate correlations (*N* = 420).

Variable	1	2	3	4	5	6	7	8	9	10
1. Age	-			-						
2. Education	0.16 ***	-		-						
3. Year of working	0.54 ***	0.07	-	-						
4. Economic constraints	0.00	−0.09	−0.13 **	-						
5. Marginalization	−0.06	−0.10	−0.07	0.56 ***	-					
6. Work volition	0.06	0.03	0.07	0.29 ***	0.17 **	-				
7. Career adaptability	0.09	0.12 *	0.14 **	−0.12 *	−0.37 ***	0.35 ***	-			
8. Decent work	0.15 **	0.15 **	0.19 **	−0.33 ***	−0.39 ***	0.15 ***	0.46 ***	-		
9. Job satisfaction	0.14 **	0.08	0.15 **	−0.20 ***	−0.35 ***	0.40 ***	0.66 ***	0.69 ***	-	
10. Life satisfaction	0.03	0.10 *	0.14 **	−0.23 ***	−0.29 ***	0.36 ***	0.43 ***	0.55 ***	0.68 ***	-
M	39.09	1.97	3.09	2.85	2.12	4.18	3.63	4.21	3.28	3.77
SD	9.30	0.67	0.88	0.91	0.72	0.75	0.63	0.86	0.67	1.27
Skewness	−0.37	2.32	−1.06	−0.11	0.24	0.15	−0.12	0.07	−0.12	−0.10
Kurtosis	0.52	0.93	−0.41	−0.50	−0.43	0.74	0.10	0.19	0.06	−0.30

* *p* < 0.5; ** *p* < 0.01; *** *p* < 0.001.

**Table 3 ijerph-19-01100-t003:** Goodness of fit statistics of measurement, hypothesized, and alternative model.

Model	*χ* ^2^	*df*	*p*	CFI	TLI	RMSEA [90% CI]
Measurement model	965.761	361	*p* < 0.001	0.92	0.91	0.063 (0.058, 0.068)
Hypothesized model	965.761	361	*p* < 0.001	0.92	0.91	0.063 (0.058, 0.068)
Alternative model	851.403	355	*p* < 0.001	0.93	0.92	0.058 (0.053, 0.063)

**Table 4 ijerph-19-01100-t004:** Bootstrap analyses of the statistical significance of the indirect effect (*N* = 420).

Path	Standardized Total Effect	Standardized Direct Effect	Standardized Indirect Effect	95% CI (Lower, Upper)
economic constraints → career adaptability	0.25 **	0.25 **	-	
economic constraints → work volition	0.37 **	0.37 **	-	
economic constraints → (work volition) (career adaptability) → decent work	−0.14	−0.33 **	0.19 **	0.10, 0.30
economic constraints → (work volition) (career adaptability) (decent work) → job satisfaction	0.02	-	0.02	−0.14, 0.18
economic constraints → (work volition) (career adaptability) (decent work) → life satisfaction	−0.03	-	−0.03	−0.19, 0.11
*social* marginalization *→* career adaptability	−0.59 **	−0.59 **	-	
*social* marginalization *→* work volition	−0.07	−0.07	-	
*social* marginalization *→* (work volition) (career adaptability) *→* decent work	−0.42 **	−0.17	−0.25 **	−0.39, −0.12
*social* marginalization → (work volition) (career adaptability) (decent work) → job satisfaction	−0.42 **	-	−0.42 **	−0.57, −0.24
*social* marginalization → (work volition) (career adaptability) (decent work) → life satisfaction	0.33 **	-	−0.33 **	−0.47, −0.16

** *p* < 0.01.

## Data Availability

The datasets analyzed for this study can be found in the Figshare Kim, MinSun (2021): decent work_data_row.sav. figshare. Dataset. https://doi.org/10.6084/m9.figshare.15505971.v1, accessed on 1 August 2021.

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
