# Peer review of "Examining Predictors and Outcomes of Decent Work among Korean Workers"

_ijerph, 2022, doi:10.3390/ijerph19031100_

Round 1

Reviewer 1 Report

I find the study interesting and well qualified in publication but with the consideration of the following concerns:

  • The introduction lacks the need for the study, the real gap the study aims to fill. Thus, I recommend including a paragraph  revealing the demand for the study.
  • After the objective of the study, main contributions should be included. In such a way, the value of the study is highlighted.
  • Limitations and future research lines should be included.

Author Response

January 1, 2021

RE: Manuscript ID: ijerph-1495346

Dear Editors:

We want to thank you for the opportunity to revise our manuscript as cited above. We have reviewed the reviewers’ comments as well as your editorial comments in the cover letter and we have made revisions to the paper accordingly. As requested, we will outline those revisions in this cover letter.

Review 1

  1. The introduction lacks the need for the study, the real gap the study aims to fill. Thus, I recommend including a paragraph revealing the demand for the study.

As Reviewer 1 suggested, we added a paragraph explaining the need for research (see lines 104 through 128 on p. 3).

  1. After the objective of the study, main contributions should be included. In such a way, the value of the study is highlighted.

As Reviewer 1 suggested, we added the contribution of the study (see lines 282 through 299 on p. 6).

  1. Limitations and future research lines should be included.

We added limitations of the study and suggestions for follow-up studies (see lines 710 on p. 16).

Reviewer 2 Report

General impressions: The authors assess a broad spectrum of possible factors that are linked to a worker’s access to decent job and the implications such access has for life satisfaction.  In relation to existing literature that has primarily focused on demographic and labor market factors, the authors of this study expand the list to include psychological, environmental, and social factors using data from Korea.  Given the scarcity of quality jobs in South Korea, a good understanding of all the possible factors that represent barriers to decent jobs is important.

General comment: It would be helpful if the authors could provide some discussion as to why it may be useful to study South Korea.  Is there anything about the country that would suggest different results?  I am also not fully convinced that measures of economic constraints are a factor (and not an outcome) of access to decent work.

Comments in more detail

Abstract:

  1. line 10: it is important to define better what is decent work
  2. line 15: Should Studies be replaced by Evidence? Studies suggest some sort of reference to existing literature rather than results in the actual paper at hand.
  3. line 20: replace “or” at the end of the line with “and”

Introduction (Section 1):

  1. line 62 (page 2): reference to comprehensive analysis of factors that affect access to decent work which necessitates a careful description of the value-added of the submitted paper
  2. line 184 on page 5: can you explain a bit better what you mean by time-selective job policy?
  3. line 189 (page 5): can you elaborate with a sentence why you think that personal and psychological factors might affect satisfaction differently in Korea?
  4. I am not sure that section 1.3. is best described as Alternative Model. To me it seems that it focuses on providing a literature review of similar studies that show evidence in support of model described in Figure 1.  Perhaps it might make more sense to introduce 1.2.4 and use as a title something that would summarize hypotheses 9 through 14 discussed in section 1.3.  I would also summarize these hypotheses in the similar way (bold type, etc.) as hypotheses 1 through 8.

Method (Section 2):

  1. line 231 (page 6): Can you please write more as to how the data were collected.  How were survey respondents selected?  What is the relevant population of interest that the sample is supposed to represent?  What was the feedback rate?  How were survey participants made aware of the survey?  Where online was the survey available?
  2. top paragraph on page 6: Can the authors comment on how the attributes of the data compare to the attributes of the working population in South Korea? Are younger workers over/under-represented? etc.
  3. in terms of section 2.2.1. it is not obvious to me that economic constrains are an outcome (and not the determinant) of access to decent work. It would be helpful to get better argument supporting the view taken by the authors.
  4. it seems that the authors do not control for demographic factors, education, experience. Is that the case?  If not, I think these factors should be included in the analysis even if they are not of primary interest (considering existing literature that documents their importance). 

Results (Section 3):

  1. It would be helpful if the authors wrote down a formula that would make it clearer how direct and indirect effect estimates were retrieved.

Discussion (Section 4):

  1. It seems that several results are inconsistent with existing evidence in related studies or in contrast with hypothesis of the authors (e.g., lines 459 482). I think the authors have to summarize for the reader how one can understand these inconsistencies – do they arise because of the sample, the measurement of x and y, the omission of relevant variables?

Author Response

January 1, 2021

RE: Manuscript ID: ijerph-1495346

Dear Editors:

We want to thank you for the opportunity to revise our manuscript as cited above. We have reviewed the reviewers’ comments as well as your editorial comments in the cover letter and we have made revisions to the paper accordingly. As requested, we will outline those revisions in this cover letter.

Review 2

  1. General comment: It would be helpful if the authors could provide some discussion as to why it may be useful to study South Korea. 

The reason why this study is necessary in Korean culture was explained in detail (see lines 104 through 128 on p. 3).

  1. I am also not fully convinced that measures of economic constraints are a factor (and not an outcome) of access to decent work.

In this study, economic constraints were viewed as a variable affecting decent jobs, and the contents were added (see lines 154 through 161 on p. 4).

  1. Abstract: line 10: it is important to define better what is decent work.

As Reviewer 2 suggested, we added a decent works definition to the abstract (see lines 12 through 14 on p. 1).

  1. line 15: Should Studies be replaced by Evidence? Studies suggest some sort of reference to existing literature rather than results in the actual paper at hand.

We revised the phrase as follows (see lines 19 through 21 on p. 1).

  1. line 20: replace “or” at the end of the line with “and”

 As you suggested, we revised I t (see lines 21 through 23 on p. 1).

  1. line 62 (page 2): reference to comprehensive analysis of factors that affect access to decent work which necessitates a careful description of the value-added of the submitted paper

We mentioned the limitations of existing studies (see lines 65 through 72 on p. 2).

  1. line 184 on page 5: can you explain a bit better what you mean by time-selective job policy?

We added a description of the time selective job as follows (see lines 229 through 240 on p. 5).

  1. Line 189 (page 5): can you elaborate with a sentence why you think that personal and psychological factors might affect satisfaction differently in Korea?

: In this study, we looked at the applicability of the PWT model, so we did not assume that a specific path would come out differently. We deleted and re-described the existing contents because they didn't seem to have been properly described (see lines 120 through 128 on p. 3).

  1. I am not sure that section 1.3. is best described as Alternative Model. To me it seems that it focuses on providing a literature review of similar studies that show evidence in support of model described in Figure 1.  Perhaps it might make more sense to introduce 1.2.4 and use as a title something that would summarize hypotheses 9 through 14 discussed in section 1.3.  I would also summarize these hypotheses in the similar way (bold type, etc.) as hypotheses 1 through 8.

Section name was modified, and hypothesis was added (see lines 246 through 301 on p. 5).

Method (Section 2):

  1. line 231 (page 6): Can you please write more as to how the data were collected.  How were survey respondents selected?  What is the relevant population of interest that the sample is supposed to represent?  What was the feedback rate?  How were survey participants made aware of the survey?  Where online was the survey available?

We added information about participants and sampling as follows (see lines 344 through 353 on p. 8).

  1. top paragraph on page 6: Can the authors comment on how the attributes of the data compare to the attributes of the working population in South Korea? Are younger workers over/under-represented? etc.

We added the information as below (see lines 309 through 313 on p. 7).

  1. in terms of section 2.2.1. it is not obvious to me that economic constrains are an outcome (and not the determinant) of access to decent work. It would be helpful to get better argument supporting the view taken by the authors.

 We assumed economic constraints as a variable affecting decent work, and added sentence (see lines 154 through 161 on p. 4).

  1. it seems that the authors do not control for demographic factors, education, experience. Is that the case?  If not, I think these factors should be included in the analysis even if they are not of primary interest (considering existing literature that documents their importance). 

As you suggested, the age, education level, and working years of the participants were added as control variables. This content was presented in the correlation analysis result table, and control variables were also indicated in the research model (see Figure 1 on p. 3, 463 on p. 10).

Results (Section 3):

  1. It would be helpful if the authors wrote down a formula that would make it clearer how direct and indirect effect estimates were retrieved.

 As you suggested, we added an example of a mediating effect formula (see lines 528 through 540 on p. 12).

Discussion (Section 4):

  1. It seems that several results are inconsistent with existing evidence in related studies or in contrast with hypothesis of the authors (e.g., lines 459 482). I think the authors have to summarize for the reader how one can understand these inconsistencies – do they arise because of the sample, the measurement of x and y, the omission of relevant variables?

We added an interpretation of the results that are inconsistent with the existing research results (see lines 576 through 614 p. 14).

Reviewer 3 Report

Authors have conducted a very detailed investigation, both theoretical and empirical. However, their results have quite a limited impact and, thus, the target audience will be reduced.

The aspects that authors may improve are related to the abstract, which lacks a logical coherence and does not state the purpose of the paper.

The sample on which they conducted the study, which should be described more detailed (for ex., for the 420 investigated workers they mentioned the mean age, which might not be relevant).

A sample description table should be included. 

Author Response

January 1, 2021

RE: Manuscript ID: ijerph-1495346

Dear Editors:

We want to thank you for the opportunity to revise our manuscript as cited above. We have reviewed the reviewers’ comments as well as your editorial comments in the cover letter and we have made revisions to the paper accordingly. As requested, we will outline those revisions in this cover letter.

Review 3

  1. The aspects that authors may improve are related to the abstract, which lacks a logical coherence and does not state the purpose of the paper.

 As you suggested, we revised the abstract (see lines 8 through 15 p. 1).

  1. The sample on which they conducted the study, which should be described more detailed (for ex., for the 420 investigated workers they mentioned the mean age, which might not be relevant). A sample description table should be included. 

 As Reviewer 3 suggested, we added the participants’ information in a table 1 (see Table 1 on p. 7).

Round 2

Reviewer 2 Report

no comments